# Prepregnancy Assessment of Liver Function to Predict Perinatal and Postpregnancy Outcomes in Biliary Atresia Patients with Native Liver

**DOI:** 10.3390/jcm10173956

**Published:** 2021-08-31

**Authors:** Nobuhiro Takahashi, Daigo Ochiai, Yohei Yamada, Masumi Tamagawa, Hiroki Kanamori, Mototoshi Kato, Satoru Ikenoue, Yoshifumi Kasuga, Tatsuo Kuroda, Mamoru Tanaka

**Affiliations:** 1Department of Pediatric Surgery, Keio University School of Medicine, Tokyo 160-8582, Japan; tkhsnbhr430@keio.jp (N.T.); kanamori06@keio.jp (H.K.); mototoshi77@keio.jp (M.K.); kuroda-t@z8.keio.jp (T.K.); 2Department of Obstetrics and Gynecology, Keio University School of Medicine, Tokyo 160-8582, Japan; ochiaidaigo@keio.jp (D.O.); akiyama_3_tohoku@yahoo.co.jp (M.T.); ikenouesatoru@me.com (S.I.); 17yoshi23.k@gmail.com (Y.K.); mtanaka@keio.jp (M.T.)

**Keywords:** pregnancy, biliary atresia, chorine esterase, MELD score, liver transplantation

## Abstract

Considering that some biliary atresia (BA) survivors with native liver have reached reproductive age and face long-lasting complications, specific attention needs to be paid to pregnant cases. This study aimed to investigate the relationship between liver function, perinatal outcomes, and prognosis. A database review was conducted to identify pregnant BA cases with native liver and perinatal data, and clinical information on BA-related complications was analyzed. Perinatal serum cholinesterase (ChE) levels, model for end-stage liver-disease (MELD) score, and platelet trends were analyzed, and the association between these indicators and perinatal outcomes was investigated. Patients were categorized into three groups according to the perinatal clinical outcomes: favorable (term babies with or without several episodes of cholangitis; *n* = 3), borderline (term baby and following liver dysfunction; *n* = 1), and unfavorable (premature delivery with subsequent liver failure; *n* = 1). Lower serum ChE levels, lower platelet counts, and higher MELD scores were observed in the unfavorable category. Borderline and unfavorable patients displayed a continuous increase in MELD score, with one eventually needing a liver transplantation. Pregnancy in patients with BA requires special attention. Serum ChE levels, platelet counts, and MELD scores are all important markers for predicting perinatal prognosis.

## 1. Introduction

Kasai portoenterostomy has been widely accepted as the primary method of surgical treatment for biliary atresia (BA), and early diagnosis and timely surgery are known to have a significant impact on long-term prognosis [1,2,3]. Owing to an improvement in clinical outcomes, an increasing number of BA patients can survive with native liver until adulthood. However, some long-term survivors of the Kasai procedure face and suffer from life-long complications, such as portal hypertension and recurrent cholangitis [4,5]. In pregnant women with native liver after BA surgery, such complications can be exacerbated by pregnancy-associated physiology [6]. Therefore, endoscopic surveillance for esophageal varices is recommended, along with the prompt initiation of antibiotic treatment for cholangitis [7].

Among pregnant patients with BA and native liver, both favorable and unfavorable perinatal courses are reported mainly because of complications, and some develop liver failure after pregnancy [6,8,9,10,11]. Sasaki et al. [9] revealed that a history of cholangitis and variceal breeding prior to pregnancy led to recurrent complications during pregnancy. Kuroda et al. [8,12] reported that the level of serum cholinesterase (ChE), which is synthesized mainly in hepatocytes and reduced in liver dysfunction, at puberty may predict pregnancy safety. To date, 58 live births in 40 pregnant BA patients have been published; however, little is known about the precise pre-pregnancy status of liver function and long-term clinical course after pregnancy [13].

In 2011, Westbrook et al. [14] demonstrated the efficacy of the model for end-stage liver disease (MELD) score in predicting outcomes in cirrhotic patients during pregnancy. In BA patients, given that the MELD score often fails to reflect the severity of BA-specific clinical symptoms, risk evaluation for pregnancy in BA patients remains unclear. Therefore, in the current study, sequential changes in the MELD score, platelet count, and serum ChE in five patients with BA were investigated during pregnancy, and their predictive efficacy for perinatal outcome and maternal prognosis are discussed.

## 2. Materials and Methods

### 2.1. Study Design

A database review was conducted to identify pregnant women who conceived after BA surgery and delivered at Keio University Hospital, between 1 April 2010 and 31 March 2020. Owing to the retrospective design of this study, opt-out consent was obtained. The study was approved by the research ethics review board of Keio University (20150103).

### 2.2. Data Collection

Antenatal data, such as maternal demographic information (age, race, parity, preexisting chronic diseases, exposure to alcohol, tobacco, and other teratogens), mode of conception, ultrasound findings, and obstetrical complications were collected retrospectively. Delivery information, including birth weight, gestational age at delivery, Apgar scores, and neonatal complications were reviewed. Additionally, we collected information on BA-related complications, including the clinical symptoms of portal hypertension (gastroesophageal varices and thrombocytopenia), coagulopathy, liver dysfunction, and cholangitis. Furthermore, biochemical data on liver function, including serum ChE level, MELD score, and platelet count during the perinatal period and after pregnancy, were collected at various periods during pregnancy. The MELD score was calculated using the following formula [15]:(1)$$\mathrm{MELD}\;=\;3.78\times log_{e}\left\{serum\;bilirubin\;\left(\mathrm{mg}/\mathrm{dL}\right)\right\}+11.2\times log_{e}\left(PT-INR\right)\;+9.6\times log_{e}\left\{serum\;creatinine\left(\mathrm{mg}/\mathrm{dL}\right)\right\}+6.43$$

## 3. Results

### 3.1. Patients

During the study period, a total of 5880 pregnant patients delivered at Keio University Hospital. Among them, five were identified as pregnant after BA surgery with native liver (Table 1).

### 3.2. Maternal Characteristics and Obstetrical Outcomes

Table 1 shows the maternal characteristics and obstetric outcomes. Perinatal outcomes after BA surgery were divided into three groups on the basis thereof: favorable (stable maternal condition with minimal complication and term baby [*n* = 3]; Patients 1–3), borderline (complication during pregnancy with subsequent worsening maternal liver function but term baby [*n* = 1]; Patient 4), and unfavorable (complication during pregnancy with subsequent deterioration of liver function and premature delivery [*n* = 1]; Patient 5).

The median gestational age at delivery was 37 weeks (range: 30–40 weeks), and the median birth weight was 2590 g (range: 842–2980 g). All patients delivered after 36 weeks of gestation, except for unfavorable patient. Three patients (Patients 2–4) developed cholangitis around the second trimester and were treated with antimicrobial agents. Two patients experienced deterioration of esophageal varices: one underwent endoscopic variceal ligation (EVL) treatment in endoscopic survey (Patient 4), and the other experienced a rupture of varices and underwent EVL (Patient 5).

### 3.3. Maternal Liver Function during the Perinatal Period

The MELD score, platelet count, and ChE were plotted during the perinatal period (Figure 1A–C). In the favorable group (Patients 1–3), maternal serum ChE levels, which are known as indicators of hepatic functional reserve, decreased during pregnancy but recovered to prepregnancy levels after each delivery. In Patient 5 (unfavorable), their low ChE level (below 200 U/L) prior to conception decreased further and did not recover to the baseline after delivery. In Patient 4 (borderline), a relatively high ChE (above 200 U/L) was seen prior to conception; however, the ChE level in this patient decreased earlier than that in the favorable group and did not return to the baseline. In the favorable group, a transient rise in the MELD score was observed, and this reflected complications such as cholangitis. The MELD score returned to the prepregnancy level after the deliveries. By contrast, a continuous uptrend in the MELD score was observed in Patients 4 and 5 after the deliveries. During the perinatal period, the platelet counts were above 10 × 10^4^/μL in the favorable group and below 10 × 10^4^/μL in Patients 4 and 5.

### 3.4. Maternal Prognosis after Delivery

In the favorable group, Patients 2 and 3 experienced a few episodes of cholangitis after pregnancy, which were successfully treated with antibiotics. Patients 1, 2, and 3 (favorable group) were in stable condition two, six, and seven years after delivery, respectively. In Patient 4 (borderline) and Patient 5 (unfavorable), liver function deteriorated gradually after delivery (Figure 1A–C). Patient 5 was listed for deceased donor liver transplantation eight years after delivery and underwent liver transplantation two years later. Unfortunately, Patient 5 died after surgery because of surgical complications. Patient 4 retained her native liver for 6.4 years after delivery with slowly deteriorating liver function.

### 3.5. Individual Clinical Courses in Pregnancy

#### 3.5.1. Patient 1: Favorable Case

A 30-year-old gravida 2, para 0 pregnant woman with no past episodes of cholangitis was referred to our hospital because of a history of BA. Fortunately, she did not experience BA-related complications, such as portal hypertension, liver dysfunction, and recurrent cholangitis. She regularly attended the preconception checkup with a pediatric surgeon and conceived spontaneously in a planned manner. At 22 weeks of gestation, gastrointestinal endoscopy did not reveal any signs of esophageal or gastric varices. Her pregnancy was uneventful until the day of delivery, and she vaginally delivered a healthy female infant weighing 2980 g at 39 weeks of gestation. Her postpartum course was uneventful, and she was in a stable condition for 1.8 year after delivery without any complications.

#### 3.5.2. Patient 2: Favorable Case

A 32-year-old gravida 1, para 0 pregnant woman was referred to our hospital because of her history of BA. She experienced an episode of cholangitis after Kasai procedure and then interrupted a routine checkup with a pediatric surgeon because her postoperative course was uneventful. She conceived spontaneously in an unplanned manner. She was diagnosed with hilar bile lake and splenomegaly but did not exhibit liver dysfunction, thrombocytopenia, or esophageal varices. Her pregnancy was uneventful, but she experienced recurrent cholangitis. Thus, she was treated with antimicrobial agents at 24 and 29 weeks of gestation. To prevent liver damage caused by repeated cholangitis, labor induction was performed at 36 weeks of gestation. She vaginally delivered a healthy female infant weighing 2578 g. Her postpartum course was uneventful. Although she experienced a few episodes of cholangitis after delivery, she maintained stable liver function for 5.8 years after the delivery. 

#### 3.5.3. Patient 3: Favorable Case 

A 39-year-old gravida 2, para 1 pregnant woman was referred to our hospital. She had at least seven episodes of cholangitis developed after 18 years old. When she was 37 years old, she delivered vaginally at 39 weeks of gestation and developed cholangitis postpartum. Although she experienced recurrent cholangitis, she did not follow the regularly attended checkup with a pediatric surgeon and conceived spontaneously in a planned manner. Her pregnancy was uneventful, but she experienced cholangitis and was treated with antimicrobial agents at 21 weeks of gestation. During pregnancy, cholangitis recurred four times, and she was treated with antimicrobial agents each time. To prevent liver damage caused by repeated cholangitis, labor induction was performed at 36 weeks of gestation, and she vaginally delivered a healthy male infant weighing 2690 g. Her postpartum course was uneventful. Cholangitis recurred at one and five years after delivery. However, her liver function remained at a good level, and she was alive for 6.8 years after her delivery.

#### 3.5.4. Patient 4: Borderline Case

A 40-year-old gravida 2, para 1 woman with a history of vaginal delivery at 39 weeks of gestation when she was 31 years old was referred to our hospital. She had experienced esophageal varices treated with EVL and some episodes of cholangitis. She started fertility treatment at her discretion and conceived spontaneously during the fertility treatment. At 25 weeks of gestation, cholangitis developed, and she required antimicrobial therapy. At 26 weeks of gestation, gastrointestinal endoscopy revealed worsening of esophageal and gastric varices, and EVL was performed. Fortunately, her pregnancy was uneventful until the day of delivery. At 37 weeks of gestation, a 2590 g healthy male infant was delivered by an elective cesarean section under general anesthesia because of low platelet count (5.3 × 10^4^/μL). During the postpartum period, pancytopenia and cholangitis were exacerbated. She is alive 6.4 years after delivery, but her liver function has gradually worsened. Liver transplantation was considered.

#### 3.5.5. Patient 5: Unfavorable Case

A 34-year-old gravida 1, para 0 pregnant woman was referred to our hospital at 28 weeks of gestation. Although she experienced several times of cholangitis, she interrupted a routine checkup with a pediatric surgeon at 29 years of age. She was diagnosed with esophageal varices at another hospital, and her condition was treated with EVL. She spontaneously conceived and underwent prenatal checkup at our hospital. At 25 weeks of pregnancy, she was transferred to our emergency room owing to rupture of esophageal varices and received EVL treatment and blood transfusion. She developed hepatic encephalopathy. At 29 weeks and 5 days of gestation, she was diagnosed with fetal growth restriction associated with oligohydramnios, and the estimated fetal weight was 900 g (−2.9 SD) with a 19 mm amniotic fluid pocket. At 30 weeks and 0 days, the patient underwent an emergency cesarean section because of non-reassuring fetal status. An 842 g male infant was delivered, with Apgar scores of four and eight at 1 and 5 min, respectively. Considering that abnormal bleeding due to coagulopathy occurred during the operation, massive blood transfusion and uterine artery embolization were required to stop the bleeding. The total volume of blood loss during delivery was 8700 mL. Her MELD score increased to 11, 13, and 21 at 1, 5, and 10 years after delivery, respectively. Eventually, she underwent liver transplantation 10 years after delivery, but she died in the perioperative period, owing to massive bleeding.

## 4. Discussion

We demonstrated the details of five perinatal courses in patients after BA surgery (Table 1), which can be summarized as follows: three term babies were successfully born with stable postpartum maternal condition (favorable), one term baby was born with worsening maternal liver function (borderline), and one premature delivery at approximately 30 weeks of gestation with subsequent maternal liver failure 8 years after delivery (unfavorable) [6,11]. 

Perinatal outcomes after BA surgery with native liver depend on prepregnancy maternal conditions, including frequent episodes of cholangitis and severity of portal hypertension manifesting gastrointestinal bleeding (Table 1) [10,11]. Cholestasis and variceal bleeding commonly occur during the second and third trimesters presumably because of high abdominal pressure, characteristic profiles of steroid hormones, expansion of maternal blood volume, and compression of the inferior vena cava [13,14]. In the current study, some patients (even in the favorable category) developed cholangitis, and their esophageal varices deteriorated; thus, BA patients with a known history of such complications before pregnancy should be followed strictly during the perinatal period.

We analyzed the potential predictors of perinatal outcomes, including the MELD score, platelet count, and serum ChE. The MELD score is an established predictor of survival in patients with liver cirrhosis. Previously, Westbrook et al. [14] demonstrated that a MELD score ≥10 was a useful predictor of significant liver-related complications in patients with liver cirrhosis during pregnancy. By contrast, a MELD score ≤6 was indicated as an assuring cutoff value. In addition, a platelet count of <11.0 × 10^4^/μL was a reliable indicator of the presence of esophageal varices in patients with cirrhosis.

The mean prepregnancy MELD scores in the favorable, unfavorable, and borderline groups were 6.9, 8.7, and 8.9, respectively. These results suggest that MELD scores between 6 and 10 may lead to unfavorable outcomes; however, further studies are warranted to obtain more accurate cutoff values. To obtain a more precise prediction, we advocate the incorporation of platelet count and ChE into our risk stratification. The presence of esophageal varices in Patients 4 and 5, which was consistent with their low platelet counts before pregnancy, corroborated the previous findings, thus leading us to suggest that low platelet count should raise awareness of the presence of esophageal varices in patients with BA. Finally, serum ChE levels <200, which indicate a significantly impaired hepatic functional reserve [16,17], were observed in Patient 5 and resulted in unfavorable perinatal outcomes. In Patient 4, a prenatal ChE >200 led to a borderline outcome as her liver function gradually worsened after pregnancy. Long-term postpartum outcome of liver function in BA patients is still unclear, and, given that the difference in MELD scores and platelet counts between Patients 4 and 5 was minimal, serum ChE may add additional information to delineate a safer pregnancy plan.

The limitations of this study include its retrospective nature, small number of patients, and absence of histological findings to corroborate liver cirrhosis. Additionally, the postpartum deterioration of liver function in Patients 4 and 5 may not be relevant to pregnancy but may reflect the natural course of BA.

Nevertheless, these findings showed the possibility that the combination of the MELD score, platelet count, and serum ChE level is useful for predicting perinatal outcomes even though the absolute cutoff values have not yet been determined, and longitudinal workup enables the delineation of the potential risks of hepatic insufficiency. If an unfavorable course is predicted beforehand, women should be informed of every possibility, including serious complications for both the mother and fetus. Early liver transplantation may be indicated in such cases. In addition, even if preconceptual prediction is favorable, strict follow-up in the perinatal period is mandatory in the pregnancy of patients with BA. 

## 5. Conclusions

In conclusion, pregnancy in patients with BA sometimes experience several types of complications and requires special attention. Serum ChE levels, platelet counts, and MELD scores have a potential to be important markers for predicting perinatal prognosis.

## Figures and Tables

**Figure 1 jcm-10-03956-f001:**
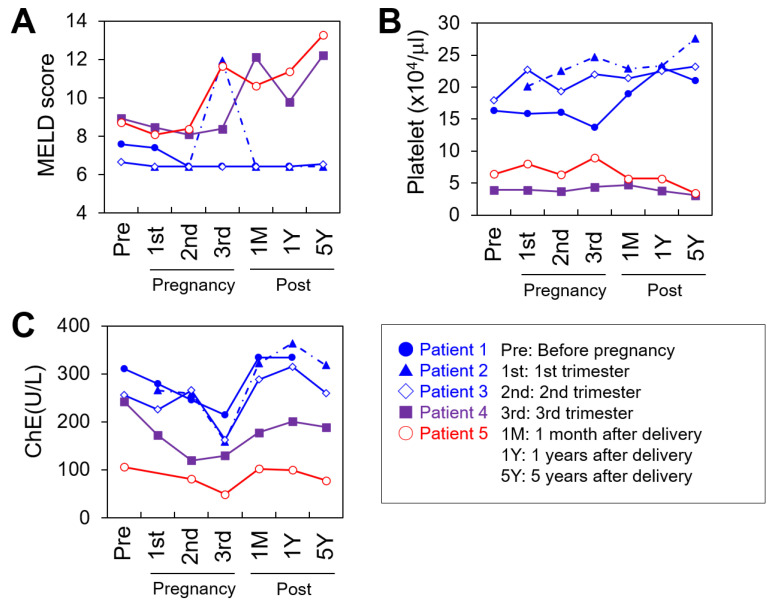
The model for end-stage liver disease (MELD) score (**A**), platelet count (**B**), and cholinesterase (ChE) plot (**C**) during the perinatal period. Pre: Before pregnancy, 1st: 1st trimester, 2nd: 2nd trimester, 3rd: 3rd trimester, 1M: 1 month after delivery, 1Y: 1 year after delivery, 5Y: 5 years after delivery.

**Table 1 jcm-10-03956-t001:** Maternal characteristics and perinatal outcomes. LTx, liver transplantation; GA, gestational age; BA, biliary atresia; VD, vaginal delivery; CS, cesarean section; GE varices, gastroesophageal varices.

Patient	Maternal Age	Nulli-Parity	Mode of Delivery	GA at Delivery (Weeks/Days)	BW	Changes in BA Complication during the Perinatal Period		
						Prepregnancy	during Pregnancy	Postpartum
1	30	Yes	VD	39 w 4 d	2980 g	None	None	None
2	32	Yes	VD	36 w 6 d	2578 g	Cholangitis	Recurrence of cholangitis	Recurrence of cholangitis
3	39	No	VD	36 w 5 d	2690 g	Cholangitis	Recurrence of cholangitis	Recurrence of cholangitis
4	40	No	CS	37 w 3 d	2590 g	Cholangitis GE varices Thrombocytopenia	Recurrence of cholangitis Exacerbation of GE varices Thrombocytopenia	Recurrence of cholangitis Exacerbation of GE varices Thrombocytopenia Liver dysfunction
5	34	Yes	CS	30 w 0 d	842 g	GE varices Thrombocytopenia Ascites	Rupture of GE varices Hepatic encephalopathy Thrombocytopenia Ascites Liver dysfunction	Exacerbation of GE varices Hepatic encephalopathy Thrombocytopenia Ascites Liver dysfunction

## Data Availability

The data presented in this study are available on request from the corresponding author. The data are not publicly available due to patients’ privacy.

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
