# Peer review of "Prepregnancy Assessment of Liver Function to Predict Perinatal and Postpregnancy Outcomes in Biliary Atresia Patients with Native Liver"

_jcm, 2021, doi:10.3390/jcm10173956_

Round 1

Reviewer 1 Report

This is an informative retrospective review of the clinical characteristics of long-term native liver survivals after KPE for biliary atresia. This study further investigated the clues for the deterioration factors before and after pregnancy. This study might add more evidence to elucidate the pathogenesis and the dynamic clinical characteristics of pregnant patients with biliary atresia. However, there are also some issues that the authors need to address:

The affiliation: Department of Pediatric surgery, Keio University

→ Department of Pediatric Surgery, Keio University

Introduction: Appropriate

Materials and Methods: Appropriate

Results:

In table 1, patient 5: GA at delivery was described as ‘3rd0d’. Please match the GA units consistently with the other cases above.

Line 99-100: All patients delivered after 36 weeks of gestation, except for unfavorable patients.

→ There was only one unfavorable patient, case 5, so it seems appropriate to write ‘patients’ as ‘patient’.

Discussion: Appropriate

Author Response

Response to Reviewer 1 Comments

We deeply appreciate critical and constructive comments from the Reviewer 1. We revised our manuscript accordingly.

1. The affiliation: Department of Pediatric surgery, Keio University Department of Pediatric Surgery, Keio University

Response: We deeply appreciate the comment. The description was corrected.

2. In table 1, patient 5: GA at delivery was described as ‘3rd0d’. Please match the GA units consistently with the other cases above.

Response: We deeply appreciate the comment. The description was corrected.

3. Line 99-100: All patients delivered after 36 weeks of gestation, except for unfavorable patients. There was only one unfavorable patient, case 5, so it seems appropriate to write ‘patients’ as ‘patient’.

Response: We deeply appreciate the comment. The description was corrected.

Reviewer 2 Report

This retrospective study reviewed pregnant biliary atresia with native liver cases and assessed liver function including serum ChE levels, platelet counts, and MELD scores to predict perinatal and postpregnancy outcomes. As the authors described, it is worth paying attention to the long-term prognosis of patients with biliary atresia as medical care advances.

The meticulous description of the interesting topic stood out. I have the following questions / comments for the authors.

1. How does your study add to the previous literature described pregnancy in patients with biliary atresia?

2. Although the number of patients included is small, is it not unreasonable to conclude that there is a relationship between the liver profile and adverse outcomes?

Author Response

Response to Reviewer 2 Comments

We deeply appreciate critical and constructive comments from the Reviewer 2. We revised our manuscript accordingly.

1. How does your study add to the previous literature described pregnancy in patients with biliary atresia?

Response: We deeply appreciate this critical and constructive comment. To date, Sasaki et al. reported the relationship between past history and perinatal complication in BA patients and Kuroda et al. reported that the possibility of ChE to predict perinatal outcome. In this study, we addressed to try to predict perinatal and postpartum clinical course by the pre-pregnant liver function. To our knowledge, this is the first trial to analysis the postpartum course including the long-term liver function at the 5 years after pregnancy and the description about long-term outcome was added to the manuscript. (line 53, and line 244-246)

2. Although the number of patients included is small, is it not unreasonable to conclude that there is a relationship between the liver profile and adverse outcomes?

Response: We appreciate the reviewer’s comment. As the reviewer pointed out, it is sure that the number of patients was too small to conclude our hypothesis. In addition to the description of the limitation (line 249-251), we modified the expression of the conclusion (line 266).

Reviewer 3 Report

Takahashi et al. address an important topic regarding pregnancy outcome in female patients with biliary atresia. 

This is a well written and interesting paper.

Minor comments:

  1. Table 1 - 5th case - there is a typo with GA at delivery. The follow up is not clear from the table - though it is described it the text, it should be better outlined in the table.
  2. You should describe the pre-pregnancy cholangitis better - when was it prior to pregnancy? how many episodes?

Author Response

Response to Reviewer 3 Comments

We deeply appreciate critical and constructive comments from the Reviewer 3. We revised our manuscript accordingly.

1. Table 1 - 5th case - there is a typo with GA at delivery. The follow up is not clear from the table - though it is described it the text, it should be better outlined in the table.

Response: We appreciate the reviewer’s comment. The description was corrected.

2. You should describe the pre-pregnancy cholangitis better - when was it prior to pregnancy? how many episodes?

Response: We really appreciate your comment. The past history of cholangitis was added to the description of each case in the results section.

Round 2

Reviewer 2 Report

I think this study is meaningful in that it delicately describes the perinatal outcomes of the Biliary atresia survivors. However, the small number of samples and the resulting difficulty in finding associations are still problematic. 
However, I think it's also meaningful to share a description of an uncommon BA survivor with the public.